# A Neural Architecture Dataset for Adversarial Robustness

## Abstract

Robustness to adversarial attacks is critical for practical deployments of deep neural networks. However, pursuing adversarial robustness from the network architecture perspective demands tremendous computational resources, thereby hampering progress in understanding and designing robust architectures. In this work, we aim to lower this barrier-to-entry for researchers without access to large-scale computation by introducing the first comprehensive neural architecture dataset under adversarial training, dubbed *NARes*, for adversarial robustness. *NARes* comprises 15,625 WRN-style unique architectures adversarially trained and evaluated against four adversarial attacks (including AutoAttack). With *NARes*, researchers can query the adversarial robustness of various models immediately, along with more detailed information, such as fine-grained training statistics, empirical Lipschitz constant, stable accuracy, etc. In addition, four checkpoints are provided for each architecture to facilitate further fine-tuning or analysis. For the first time, the dataset provides a high-resolution architecture landscape for adversarial robustness, enabling quick verifications of theoretical or empirical ideas. Through *NARes*, we offered some new insight and identified some contradictions in statements of prior studies. We believe *NARes* can serve as a valuable resource for the community to advance the understanding and design of robust neural architectures.

## 1 Introduction

Robustness to adversarial attacks is essential for the reliable deployment of deep neural networks in real-world applications. In the quest for effective defenses, much of the existing research has concentrated on enhancing adversarial training (AT) techniques (Madry et al., 2018; Zhang et al., 2019; Wang et al., 2020; Rice et al., 2020). These methods have been predominantly explored within the confines of variants of wide residual networks (WRNs) (Zagoruyko & Komodakis, 2017). Despite the pivotal role that novel network architectures have played in the broader success of deep learning (He et al., 2016; Dosovitskiy et al., 2021; Brown et al., 2020), advancements in enhancing adversarial robustness (AR) through architectural innovations remain limited. Nonetheless, a growing body of empirical evidence suggests a significant correlation between network architecture and adversarial robustness (Huang et al., 2021; 2023; Peng et al., 2023). This observation underscores the urgent need for a comprehensive investigation into how different network architectures can contribute to improving adversarial robustness. We posit that such a large-scale exploration is both timely and critical.

**Limitation of current architecture datasets for AR**. Unfortunately, a comprehensive evaluation of network architectures for AR requires tremendous computation, imposing a steep barrier-to-entry on researchers without access to large-scale resources. To facilitate AR research on network architecture while circumventing the aforementioned issue, two neural architecture (NA) datasets for AR have been proposed (Jung et al., 2023; Wu et al., 2024). There are three main limitations of these two existing datasets: ① Both datasets adopt the micro architecture search space proposed in NAS-Bench-201 (Dong & Yang, 2019) that solely concerns the topological design of a cell that is repeated many times to form an architecture. However, most theoretical and empirical studies of AR with architecture design were conducted on the macro search spaces, specifically WRN-style architectures, leaving a gap between these datasets and other research. ② The network models from both datasets are small-scale architectures with number of parameters ranging between 0.07M∼1.53M and contains many incapable failure models, which do not satisfy the high-capacity demand for AR. ③ Neither

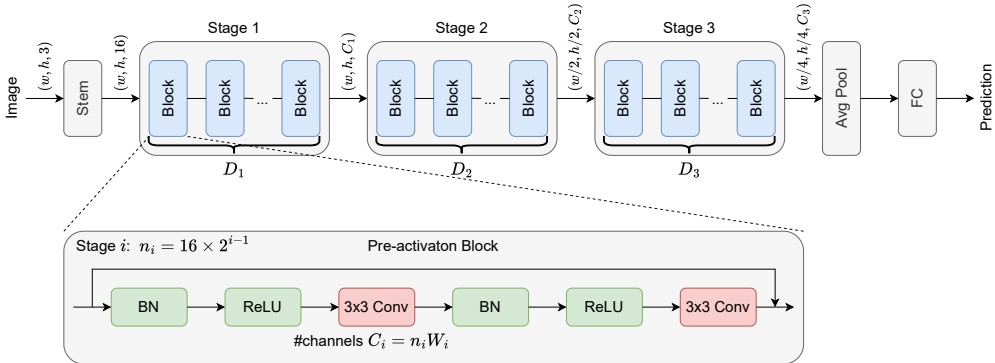

Figure 1: Design space of *NARes*. It explores wide residual networks with different depth and width settings. The encoding scheme adopts a 6-dimension vector $[D_1, W_1, D_2, W_2, D_3, W_3]$, where $D_{i \in \{1,2,3\}} \in \{4, 5, 7, 9, 11\}$ is the depth and $W_{i \in \{1,2,3\}} \in \{8, 10, 12, 14, 16\}$ is the width factor for each stage. We use the pre-activation design for every block with two $3 \times 3$ convolution layers.

dataset provided informative metrics along the training process, which are crucial for understanding how adversarial training affect AR via architecture designs; and neither dataset evaluated architectures against AutoAttack, which is currently the most reliable metric of AR.

**Advantages of *NARes*.** To address the above issues, we propose a new NA dataset focusing on the macro search space based on WRN with varying depths and widths, as shown in Figure 1. The 15,625 architectures ($2.4\times$ larger than existing NA datasets for AR (Jung et al., 2023; Wu et al., 2024)) in our design space span a wide spectrum of model capacities, i.e., with the number of parameters from 23.25M to 266.80M. In contrast to existing NA datasets, all of our models are considered applicable to the AR scenario. We explicitly mitigate robust overfitting during the training through an independent validation set. Moreover, we provide a richer set of evaluation metrics than the above NA datasets. In addition to accuracies on four adversarial attacks, including AutoAtack, we provide diagnostic information like stable accuracy and empirical Lipschitz constant (LIP) under attacks, which would help develop insights into network architecture designs for AR.

**Key takeaways:** According to analysis on *NARes*, we have several key findings: ① Compared to parameters, increasing MACs budget is preferred for AR. ② Stable accuracy consistently indicates the corresponding AR, while lower LIP is a necessary condition for AR. Increasing the depth at last stage will statistically decrease the LIP. ③ Statements in previous principle designs might not be reliable. For example, reducing the last stage capacity will result in a statistical decrease in AR; and previous robust neural architecture principles Huang et al. (2023); Peng et al. (2023) cannot correctly depict the optimal architectures. ④ Every depth and width values collectively determines the AR of the model, folding them into one dimensional variable might not be sufficient.

**NAS benchmark for adversarial robustness.** *NARes* can also serve as a dataset for the NAS community, which opens the door for easily exploring macro search spaces on AR. Since prior NAS methods on AR primarily focused on the topology of cell-based (micro) search space, we expect *NARes* to encourage more focus on macro search spaces of architectures and bridge the gap to other AR investigation areas.

We summarize the primary contributions of *NARes* as below:

1. **The first large-scale NA dataset on the macro search space**. *NARes* adversarially trained 15,625 architectures and evaluated them against AutoAttack along with three additional white-box attacks and 19 common corruptions, requiring a total of 44 GPU years to build.
2. **Insights for the future AR research from architecture angle**. Based on *NARes*, we have a deeper insight into how architectures affect the AR of models as mentioned above. *NARes* provides an opportunity to validate old and new ideas freely, contributing to the harmony between the theoretical and empirical studies on AR with respect to architecture. Besides, it serves as a time-free NAS benchmark on the macro search space, advocating new advanced searching algorithms.

3. **Assessable and reproducible model weights and AR evaluation**. We will open-source the training and evaluation code of *NARes*, along with 62,500 pre-trained checkpoints (four per architecture) to foster further development, analysis of neural architectures on AR.

## 2 RELATED WORK

### 2.1 ADVERSARIAL EXAMPLE AND DEFENSE

The vulnerability of deep neural networks (DNNs) on adversarial examples (AEs) was first studied in Szegedy et al. (2014), where AEs are crafted inputs that trick the model into outputting incorrect answers.

**Adversarial attacks.** White-box attacks utilize DNN models' internal information, such as gradients, to iteratively adjust the AEs, with noticeable methods including FGSM (Goodfellow et al., 2015), PGD (Madry et al., 2018), and CW (Carlini & Wagner, 2017). These methods progressively perturb a clean image $x$ along the direction of the gradient of a loss function $L$ on $x$, and the perturbations are restricted within a small neighborhood $\mathbb{B}(x, \epsilon)$: $\hat{x}_{t+1} = \Pi_{\mathbb{B}(x,\epsilon)}[\hat{x}_t + \alpha \cdot \text{sign}(\nabla_x L(\hat{x}_t, y))]$, where $\Pi_{\mathbb{B}(x,\epsilon)}$ projects the perturbed image back to $\mathbb{B}(x, \epsilon)$, i.e., an $\ell_p$-ball with radius $\epsilon$ around $x$. The corresponding accuracy on AEs under white-box attacks can be deemed a type of worst-case analysis for the robustness of neural network models. Recent advances in adversarial attacks include AutoAttack (AA) (Croce & Hein, 2020a), which uses PGD with adaptive step size and aggregates multiple attacks. Despite being computationally expensive to execute, AA has been widely used for benchmarking adversarial robustness (Croce et al., 2021).

**Adversarial training (AT) as the de-facto defense.** The main idea of AT is to add AEs to the training set to enhance the robustness of the DNN models against adversarial attacks. It was first proposed by Goodfellow et al. (2015) and widely adopted after Madry et al. (2018). Generally, AT can be formulated as a min-max optimization problem, where the training algorithm minimizes the loss on AEs, which is maximized by the inner attack algorithm. This motivated a series of works to improve AT, including ALP (Kannan et al., 2018), TRADES (Zhang et al., 2019), and MART (Wang et al., 2020). In addition, AT can be combined with other defense mechanisms, such as early stopping (Rice et al., 2020) for robust overfitting, weight ensembling (Izmailov et al., 2018; Chen et al., 2021; Wang & Wang, 2022), and data augmentation (Rebuffi et al., 2021b) or external data through generative modeling (Gowal et al., 2021; Sehwag et al., 2022; Wang et al., 2023).

### 2.2 EXISTING INVESTIGATIONS ON NEURAL ARCHITECTURES FOR ADVERSARIAL ROBUSTNESS

An orthogonal group of methods seek adversarial robustness from the perspective of network architectures. Existing works can be classified into two categories: (1) manual design through architectural insights and (2) automated design through NAS.

In the first category, existing efforts primarily focus on wide residual networks (WRNs) (Zagoruyko & Komodakis, 2017) and have attempted to empirically derive design principles based on WRN architectures that are robust against adversarial attacks. RobustWRN built a connection between the AR loss and the model's Lipschitz constant and observed that reducing the depth and width at the last stage leads to more robust WRNs (Huang et al., 2021); Huang et al. (2023) found that deep but narrow residual networks are adversarially more robust than wide but shallow networks; RobustPrinciple further refined the principles and proposed a range of effective depth and width ratios for robust WRNs (Peng et al., 2023). However, these design principles were derived from a limited number (e.g., a few hundred) of sampled architectures, where the landscape of the architecture space has not been exhaustively explored. Therefore, these design principles might not be optimal and potentially biased due to randomness in sampling architectures.

Moreover, there is the disharmony among these studies. Empirical studies (Xie & Yuille, 2019; Madry et al., 2018) of design principles have shown that AR demands higher model capacity (width and depth) than traditional training, and Madry et al. (2018) explained that higher model capacity would help construct a more complicated decision boundary for robustness. However, there are disagreements in theoretical analysis. On the one hand, recent works suggested that over-parameterization might hurt the robustness (Gao et al., 2019; Wu et al., 2021; Huang et al., 2021;

Hassani & Javanmard, 2024; Zhu et al., 2022); on the other hand, some works argued that enough parameters are essential to guarantee robustness (Bubeck & Sellke, 2021; Bubeck et al., 2021). Some of these theoretical analyses rely on specific lazy training initialization and additional assumptions or are limited to two-layer networks (Zhu et al., 2022), which might not be well generalized to the real models. We hope *NARes* will help eliminate this dilemma.

Alternatively, NAS algorithms automate the process of designing robust architectures by searching in a design space. The search algorithms include differential optimizations (Mok et al., 2021; Hosseini et al., 2021), evolutionary algorithms (Kotyan & Vargas, 2020) and random search (Guo et al., 2020). Compared to traditional NAS, new objectives or structures for robustness are incorporated during the search. However, in this category, the search space primarily consists of cell-based architectures, which focus on the topology of the architecture. In contrast, macro architectural search spaces such as the widths and depths of WRN have not been fully investigated. However, many theoretical and empirical studies on AR with architecture design were conducted on the WRN search space, leaving a significant gap between NAS and the AR community.

### 2.3 EXISTING NEURAL ARCHITECTURE DATASETS AND BENCHMARKS FOR MODEL ROBUSTNESS

There is a growing interest in searching for robust architectures. As such, two NAS datasets on robustness have been proposed recently. Jung et al. (2023) reused weights from NAS-Bench-201 (Dong & Yang, 2019) and evaluated the models' robustness in that cell-based search space. The robustness of common corruptions and several adversarial attacks under different maximum perturbations was evaluated and studied. However, these models were learned through standard training. Wu et al. (2024) resolved this concern by training all 6466 non-isomorphic models with adversarial training and extended experiments to three image datasets. Nonetheless, as discussed in Sec. 1, several limitations remain.

Besides these two datasets that focusing on a family of homogeneous architectures, there are also works on benchmark existing models with various heterogeneous architectures. Tang et al. (2021) benchmarked 49 architectures based on human-designed networks and 1200+ subnet architectures from NAS, including state-of-the-art CNN models, Vision Transformers and MLP-Mixer. Li et al. (2023) proposed a benchmark of AR under distribution shift, where 706 robust models under various architectures were tested. In this work, we focus on a type of homogeneous WRN architectures and attempt to thoroughly explore the architecture space for adversarial robustness.

## 3 *NARes*: A LARGE-SCALE NA DATASET UNDER AT

### 3.1 DESIGN OF *NARes*

**Design Space:** As illustrated in Fig. 1, we use the wide residual network (WRN) (Zagoruyko & Komodakis, 2017) as the fundamental architecture of *NARes* and explore different depth and width settings in each stage. The architecture comprises three stages, with each stage stacking multiple blocks, each consisting of two $3 \times 3$ convolution layers. The input is downsampled at the second and third stages by the first convolution layer with a stride of 2. Additionally, each block uses a pre-activation design for better robustness (Huang et al., 2023). The encoding scheme adopts a 6-dimension vector $[D_1, W_1, D_2, W_2, D_3, W_3]$. $D_{i \in \{1,2,3\}} \in \{4, 5, 7, 9, 11\}$ is the number of blocks in each stage. $W_{i \in \{1,2,3\}} \in \{8, 10, 12, 14, 16\}$ is the width factor which controls the number of channels $n_i W_i$ at the block of stage $i$, with $n_i = 16 \times 2^{i-1}$. In summary, there are $5^6 = 15625$ different architectures, including many models that are commonly employed in adversarial robustness research, such as WRN-34-10 ($D_{i \in \{1,2,3\}} = 5$, $W_{i \in \{1,2,3\}} = 10$), and WRN-70-16 ($D_{i \in \{1,2,3\}} = 11$, $W_{i \in \{1,2,3\}} = 16$).

**Training Setting:** A fixed set of hyperparameters was used for training all models in *NARes*. Every model was trained with the standard adversarial training (AT) by Projected Gradient Descent (Madry et al., 2018), for 100 epochs on the full CIFAR-10 training set (Krizhevsky, 2009). The learning rate decayed by a factor of 0.1 at the epoch 75 and 90. To avoid the Robust Overfitting (Rice et al., 2020) during the later training stage of AT, we applied the early stopping strategy by recording the best PGD-CW[40] accuracy (see Sec. 3.2) on a separate validation set. Other training settings are detailed

Table 1: Details of Data in *NARes*

| | |
|---|---|
| **Per Epoch** | Adversarial training loss and accuracy |
| | Validation loss, clean accuracy |
| | Validation accuracy of PGD[20] and PGD-CW[40] under $\ell_\infty$ |
| | Corresponding stable accuracy and empirical Lipschitz constant |
| **Per Architecture** | Number of Parameters (#Params) and Number of MACs (#MACs) |
| | Test loss and clean accuracy [*] |
| | Test accuracy of FGSM, PGD[20] and PGD-CW[40], AA-Compact under $\ell_\infty$ [*] |
| | Test stable accuracy and Empirical Lipschitz constant of PGD[20] and PGD-CW[40] [*] |
| | Accuracies and losses under common corruptions in CIFAR-10-C [*†] |
| | Four checkpoints of weights at the epoch 74, 89, 99 and the best epoch |

[*] : Evaluating the best checkpoint of an architecture.
[†] : Including 19 corruption types under 5 severity levels.

in Appendix B. Through the AT process, we saved four checkpoints: two before the learning rate decay (the epoch 74 and 89), the last epoch, and the best epoch based on the PGD-CW[40] accuracy. We exhaustively trained all 15625 model architectures in the design space, with the entire training process costing approximately 13.1K GPU days ($\sim$ 36 GPU years).

## 3.2 METRICS AND DIAGNOSTIC INFORMATION

During the training of each network architecture, we logged the adversarial training loss and accuracy for every epoch. After each training epoch, we used CIFAR-10.1 (Recht et al., 2018), a dataset with 2K images sampled by the similar creation process as CIFAR-10, as the validation set and evaluated the model's clean accuracy and accuracy against two attacks: PGD[20] and PGD-CW[40]. The PGD[20] attack (Madry et al., 2018) uses a random start and applies 20 steps with step size $0.8/255$ and maximum $\ell_\infty$ perturbation $\epsilon = 8/255$. The PGD-CW[40] attack applies 40 steps with the Carlini-Wager loss [1] and keeps the other setting as PGD[20].

Besides adversarial accuracies on the validation set after each epoch, we also evaluated each corresponding attack's stable accuracy and empirical Lipschitz constant (Yang et al., 2020; Huang et al., 2021). The stable accuracy measures the perturbation stability of the model, calculated by measuring whether the adversarial attack can change its prediction: $\| \{x \sim \mathbb{D}_{val} : f_\theta(x) = f_\theta(\hat{x})\} \| / \|\mathbb{D}_{val}\|$, where $\hat{x}$ is the AE of $x$ after attack on the validation set $\mathbb{D}_{val}$. The empirical Lipschitz constant measures the model's local Lipschitz constant within the attack's perturbation range $\mathbb{B}(x, \epsilon)$, which reflects the model's maximum output changes in a small input perturbation and is directly related to the adversarial training loss (Wu et al., 2021). We estimate it by

$$L(\mathbb{B}, \epsilon) = \frac{1}{\|\mathbb{D}_{val}\|} \sum_{x \in \|\mathbb{D}_{val}\|} \frac{\|f_\theta(x) - f_\theta(\hat{x})\|_1}{\|x - \hat{x}\|_\infty} {}_2. \tag{1}$$

After training, we evaluated the clean accuracy and adversarial robustness on the CIFAR-10 test set at the best epoch of each architecture. We consider the FGSM (Goodfellow et al., 2015), PGD[20], and PGD-CW[40] attacks on the $\ell_\infty$-norm perturbation with step size $0.8/255$ and $\epsilon = 8/255$. Besides, their stable accuracy and empirical Lipschitz constant were also recorded. We also evaluated the robustness against a compact version of AutoAttack (Croce & Hein, 2020b) with $\epsilon = 8/255$, which consists of untargeted and targeted APGD. We denote it as *AA-Compact*. It helps to reduce the

---

[1]The untargeted version of the original loss used in Carlini & Wagner (2017): $L_{\text{PGD-CW}}(\hat{x}_k) = -\max([\boldsymbol{Z}(\hat{x}_k)_t - \max_{i \neq t} \boldsymbol{Z}(\hat{x}_k)_i], 0)$, where $\boldsymbol{Z}(\hat{x}_k)$ is the logits of the model on the perturbed image $\hat{x}_k$ at attack step $k$ and $t$ is the true label of the original image $x$.

[2]We follow the practical implementation of Huang et al. (2021) in `https://github.com/HanxunH/RobustWRN`, using the attack samples to replace the original maximum operation.

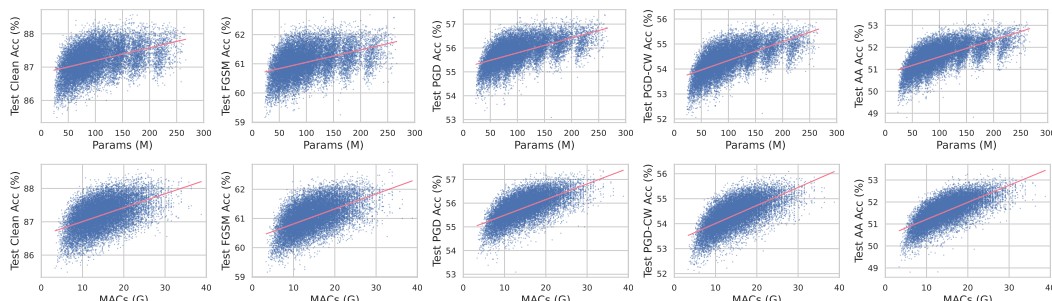

Figure 2: The clean accuracy and adversarial accuracies under different attacks on models in *NARes*. Specifically, for each architecture, we select the best model based on the PGD-CW[40] accuracy of the validation set and evaluate it on the test set. The clean accuracy and FGSM, PGD[20], and PGD-CW[40] accuracy on the test set are reported.

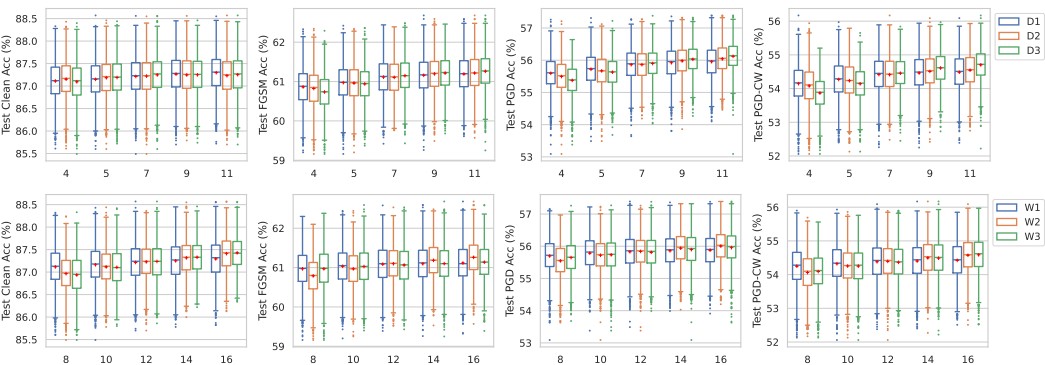

Figure 3: The distribution of clean accuracy and adversarial accuracies under different depth and width settings. The accuracies are evaluated on the test set, and the red "+" sign represents the mean accuracy of each group.

expensive computational evaluation cost, and previous works (Rebuffi et al., 2021a), along with our experiments in Table. 3, have shown a good approximation to the AutoAttack.

We also evaluated the best models' robustness on common corruptions to complement the metrics for adversarial robustness on CIFAR-10-C (Hendrycks & Dietterich, 2018) dataset, which contains 19 diverse corruption types in nature. Each corruption type contains 10K labeled images under five severity levels, perturbed from the test set of CIFAR-10. Finally, every architecture's number of parameters and MACs were recorded as metrics for model complexity.

The entire evaluation costs 2.9K GPU days ($\sim$ 8 GPU years). In summary, *NARes* offers the following information for each architecture in the above design space in Table 1, providing a comprehensive dataset for model robustness from the network architecture perspective:

## 4 STATISTICS OF *NARes* DATASET

This section overviews the statistics of *NARes* on AR. We demonstrate the model's AR metrics and their relationship to stable accuracy and empirical Lipschitz constant. Then we validate the statements in previous robust architectural design principles and explore the features of promising architectures within *NARes*. Extended analyses are detailed in Appendix A.

### 4.1 ROBUST ACCURACY

To explore various architecture designs of WRN, we analyze the clean accuracy and adversarial accuracies of the aforementioned attacks on the test set. Fig. 2 compares the test accuracies with

the number of parameters and MACs (#Params and #MACs), respectively. Our major discovery is that under the search space of *NARes*, the upper bound of the AR will quickly meet the bottleneck by increasing #Params. However, the lower bound will consistently benefit by increasing it. This reveals a complex relationship between the model size and AR. When the model complexity is increased by #MACs, we observe a more obvious trend, where both the upper and lower bounds of accuracies are improved. Although the relationship between #Params and AR was primarily studied, this observation suggests that increasing the budget on MACs is preferred to enhance robustness than the parameter budget.

Moreover, we examine the effect of the single depth or width in our decision vector. The results are shown in Fig. 3. We find that increasing any single value of the depth or width factor will boost the clean accuracy and adversarial robustness from the model distribution perspective, contradicting the previous consensus that the model capacity at the last stage should be kept small (Huang et al., 2021; Peng et al., 2023). We explain it as a consequence of low empirical Lipschitz constant discussed in Sec. 4.2, where large width and depth settings could also result in low Lipschitz constant, leading to high AR.

## 4.2 Stable Accuracy and Empirical Lipschitz Constant

Wu et al. (2021) found that the robust examples can be divided into two overlapping groups: correctly classified examples and stable examples; and compared to clean accuracy, stable accuracy is strongly correlated to AR. In Fig. 4, we plot the test stable accuracy of $PGD^{20}$ in *NARes* and its distribution under different depth and width settings. As stated in previous works (Wu et al., 2021; Huang et al., 2021), the stable accuracy is approximately correlated to the AR. Besides, we observe that increasing the depth at each stage would also consistently improve the stability. However, increasing width would only benefit a trivial stability at stages 1 and 2, and could cause a slight drop at stage 3.

Besides stable accuracy, the local Lipschitz constant was utilized to establish a theoretical connection between AR and model architecture. In summary, there is a trade-off between model capacity (width and depth) and the local Lipschitz upper bound, where the latter is directly related to the adversarial training loss (Wu et al., 2021). Reducing it would improve the perturbation stability. A consensus of AR was to reduce the model capacity of the last stage (Huang et al., 2021; Peng et al., 2023). Therefore, we further explore the empirical Lipschitz constant (LIP) on the test set with $PGD^{20}$, as shown in Fig. 5. Overall, in contrast to the trend for stable accuracy, the relationship between the LIP and AR is complex. Nevertheless, models with high AR indeed have low LIP, suggesting a relatively small LIP is a necessary condition for high robustness. For the effect of single decision variable, unlike the predictions from previous theoretical analysis (Gao et al., 2019; Wu et al., 2021; Huang et al., 2021; Hassani & Javanmard, 2024; Zhu et al., 2022), there is no clear evidence that LIP grows with the increase of depth or width. Surprisingly, increasing the depth at the last stage would statistically decrease the model's LIP. Meanwhile, the LIP is less sensitive to the width factor at all stages.

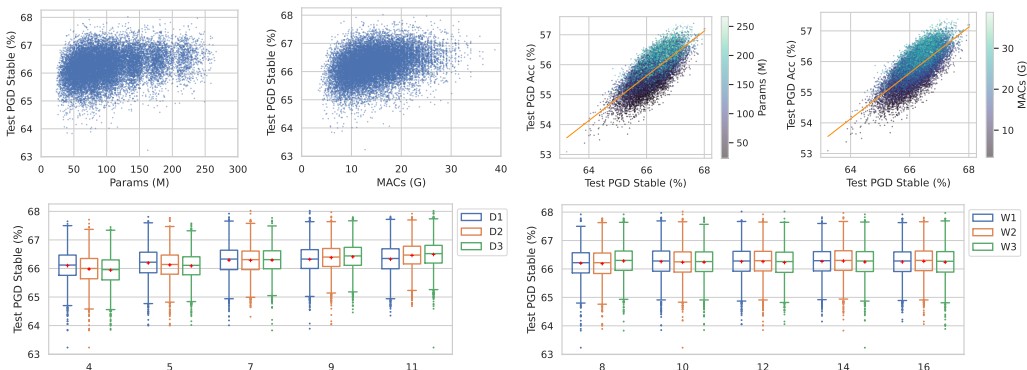

Figure 4: The statistics of $PGD^{20}$ stable accuracy on the test set. In box plots, the red "+" sign represents the mean accuracy of each group.

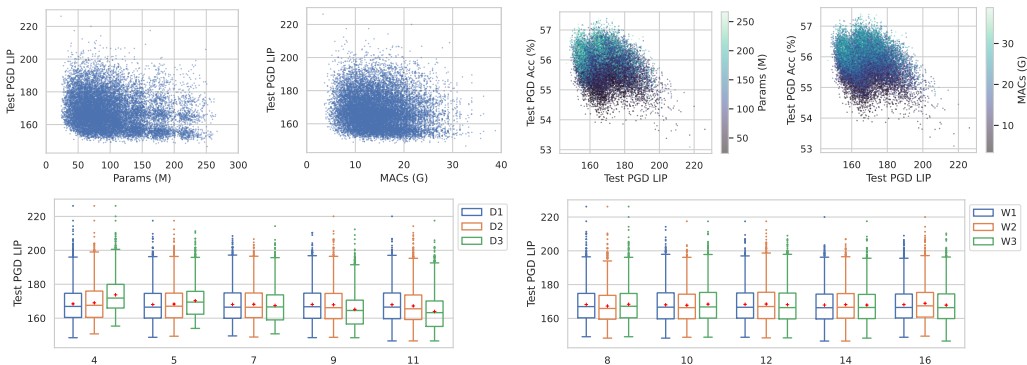

Figure 5: The statistics of PGD$^{20}$ empirical Lipschitz constant (LIP) on the test set. In box plots, the red "+" sign represents the mean accuracy of each group.

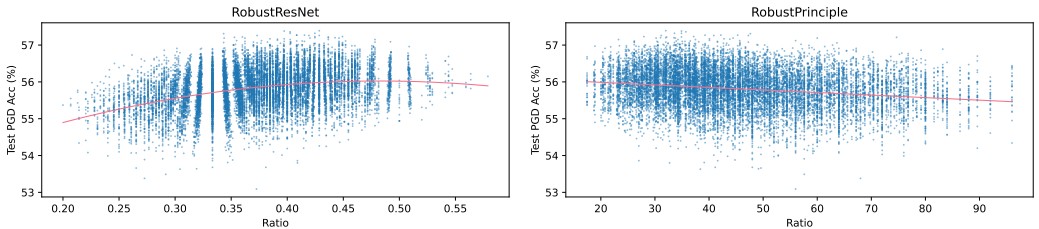

Figure 6: Model distribution on the test PGD$^{20}$ accuracy and the depth-width ratio proposed in RobustResNet and RobustPrinciple.

### 4.3 VALIDATING PREVIOUS ROBUST ARCHITECTURE PRINCIPLES

*NARes* provides the architecture landscape on robustness with high resolution, so we can easily validate the correctness of statements in previous robust architecture principles which were also based on this search space. In Sec. 4.1, we have already found that decreasing the depth and width at the last stage is not statistically beneficial for AR.

Then, we validate the statements proposed in recent works in Fig. 6. RobustResNet (Huang et al., 2023) utilized a fixed depth-width ratio, where $r_{RobustResNet} = \sum_{i \in \{1,2,3\}} D_i / (\sum D_i + \sum W_i)$ has an optimal value for AR. We plot the distribution of models concerning this ratio on the test PGD$^{20}$ accuracy. Although we can fit a quadratic regression curve, the PGD$^{20}$ accuracy falls into a wide range under a similar ratio. Therefore, the ratio only gives a coarse architectural manual for AR. Similarly, we plot the depth-width ratio $r_{RobustPrinciple} = \frac{1}{2}(C_1/D_1 + C_2/D_2)$ from RobustPrinciple[3] (Peng et al., 2023), which assumes that AR is negatively proportional to the ratio. The results demonstrate that, although there indeed is a vague tendency following the assumption, using a fixed range of depth-width ratio is also considered a coarse architecture guideline.

In summary, the above validation exposes the potential limitation of previous empirical studies with limited samples. With the real and informative metrics in *NARes*, we can provide a more comprehensive and accurate understanding of the robust architecture design principles.

### 4.4 PROMISING ARCHITECTURES

Besides the analysis of a single architecture variable in Sec. 4.1, we believe any choice on a single depth or width will not be a deterministic factor for the robustness, and the model's robustness is collectively determined by all factors in the decision vector, i.e., $[D_1, W_1, D_2, W_2, D_3, W_3]$. To explore the intrinsic relations among depths and widths for promising robust architecture under different model complexity budgets, we calculate the Pareto rank based on the test PGD$^{20}$ accuracy

---

[3]$C_i$ is the number of channels at stage $i$, as shown in Fig. 1.

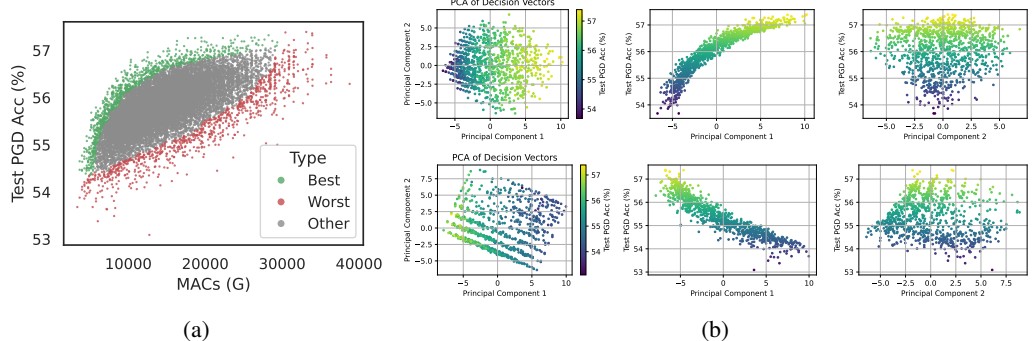

(a)                                                                      (b)

Figure 7: The relation of decision vector on robustness. **(a)**: The selected best and worst models based on the Pareto rank of test PGD accuracy and #MACs. **(b)**: The PCA(n=2) results on the best (Top) or worst (Bottom) models' decision vector and their projections.

and #MACs and select models with rank smaller than 16 as the best architectures. Similarly, we get the worst architectures with inverse Pareto rank. The selection results are shown in Fig. 7a. Since the PGD[20] accuracy contains noise, we further apply Principal Component Analysis (PCA) on the decision vectors of best and worst samples respectively. The PCA results are shown in Fig. 7b. We find that robustness is only highly correlated with the projection on the first component of PCA. The corresponding principal component $[0.378, 0.315, 0.350, 0.517, 0.441, 0.416]$ and $-[0.283, 0.282, 0.468, 0.537, 0.454, 0.356]$ represent a denoised linear relationship among depths and width for the best and worst set of models. It substantiates our statement that each architecture variable is equally important for AR, and principles that folds decision vector into a single variable like the depth-width ratio are not sufficient.

We emphasize that this linear combination is not the direct advice for new architectural principles by scaling models along the best models' PCA direction, since this conclusion is derived and confined to our search space. The above selection mechanism lets the best and worst set of models intersect at the two sides of the model complexity range. Therefore, scaling models up or down that is beyond our search space range will no longer guarantee the new models fall into the real best set.

## 5  *NARes* AS A NAS BENCHMARK

In this section, we demonstrate another application of *NARes*: as a NAS benchmark dataset. We consider several black-box NAS algorithms as the baseline algorithms, including Random Search (Li & Talwalkar, 2019), Local Search (White et al., 2021b), Regularized Evolution (RE) (Real et al., 2019) and BANANAS (White et al., 2021a). The objective is to find an architecture that maximizes the PGD[20] accuracy on the validation set at its best epoch, with a maximal 500 queries (3.2% of the search space size). Then, the metrics of the best architecture during the search are reported. The detailed experiment settings are discussed in Appendix C.1. All algorithms are independently tested over 400 runs, and the average results are listed in Table 2.

The results demonstrate that RE and BANANAS achieve better performance than classical search algorithms in *NARes*. Specifically, BANANAS achieves the best performance on the search objective and other validation accuracies and is more stable than other algorithms. This suggests that advanced search techniques are indeed helpful in our search space. Moreover, the robustness of the test set shows that both RE and BANANAS search for an architecture with similar high robustness.

## 6  CONCLUSION AND FUTURE WORK

This paper presents *NARes*, a new neural architecture dataset for adversarial robustness, which contains weights and robustness metrics on 15625 unique models based on wide residual networks (WRNs). This is the first dataset that exhaustively evaluated different depth and width settings on a macro search space for adversarial robustness. According to the analysis in Sec. 4, we have found some deep architectural insights, some of which may challenge previous statements. In the future,

Table 2: Results of different NAS algorithms on *NARes*. The algorithms search the best architecture based on the PGD[20] accuracy on the validation set, and the mean and the standard variance of robustness metrics on the best architecture are reported over 400 runs.

| Accuracy | Optimal[*] | Random Search | Local Search | RE | BANANAS |
|---|---|---|---|---|---|
| Val Clean | 78.25 | $75.88 \pm 0.56$ | $75.84 \pm 0.59$ | $76.07 \pm 0.39$ | $\mathbf{76.10 \pm 0.38}$ |
| Val PGD[20†] | 38.80 | $38.18 \pm 0.22$ | $38.17 \pm 0.22$ | $38.50 \pm 0.24$ | $\mathbf{38.55 \pm 0.24}$ |
| Val PGD-CW[40] | 37.55 | $36.58 \pm 0.38$ | $36.60 \pm 0.41$ | $36.96 \pm 0.42$ | $\mathbf{36.99 \pm 0.40}$ |
| Test Clean | 88.57 | $\mathbf{87.28 \pm 0.37}$ | $87.26 \pm 0.39$ | $87.24 \pm 0.30$ | $87.22 \pm 0.28$ |
| Test FGSM | 62.68 | $61.38 \pm 0.34$ | $61.39 \pm 0.36$ | $\mathbf{61.46 \pm 0.25}$ | $61.45 \pm 0.23$ |
| Test PGD[20] | 57.39 | $56.44 \pm 0.36$ | $56.47 \pm 0.37$ | $\mathbf{56.68 \pm 0.29}$ | $\mathbf{56.68 \pm 0.26}$ |
| Test PGD-CW[40] | 56.17 | $54.86 \pm 0.38$ | $54.91 \pm 0.39$ | $\mathbf{55.06 \pm 0.26}$ | $55.05 \pm 0.24$ |
| Test AA[‡] | 53.48 | $52.18 \pm 0.39$ | $52.24 \pm 0.39$ | $\mathbf{52.45 \pm 0.28}$ | $\mathbf{52.45 \pm 0.25}$ |
| Test Corruption | 80.22 | $78.89 \pm 0.36$ | $78.90 \pm 0.36$ | $78.90 \pm 0.24$ | $\mathbf{78.91 \pm 0.22}$ |

[*] : "Optimal" refers to the highest achievable accuracy in the dataset of *NARes*.
[†] : The objective for NAS.
[‡] : We use AA-Compact, a compact version of AA.

we hope this dataset will continually contribute to the development of adversarial robustness in neural architectures, both empirically and theoretically. For the neural architecture search (NAS) community, *NARes* lowers the barriers to entry and bridges the gap to other adversarial robustness research. Theories on NAS of robustness might benefit from it and derive new algorithms.

### 6.1 LIMITATIONS

Adversarial training and evaluation require substantial computational resources. Consequently, our dataset currently includes only a single sweep of the entire search space, which may introduce some noise into each architecture's data. To migrate the noise, we handle architectures from a distribution perspective (Sec. 4), rather than focusing on specific architectures. And we recommend that future analyses on *NARes* consider network design spaces with statistical tools (Radosavovic et al., 2019; 2020). For the same reason on computational cost, the dataset currently is built on CIFAR-10, which may limit the generalization of the findings. Therefore, we recommend using *NARes* as the first step of finding new insights or as a quick verification of some new ideas, which will massively reduce the time cost. Then, the findings can be further validated on other datasets under a few experiments, which will finally help the development of new robust architectures. In addition, our search space may not encompass all WRN architectures that also fit within our #Params or #MACs range, leaving some gaps in the comprehensive overview of architecture design concerning model complexity budgets. Lastly, the accuracy correlation between the validation and test sets is relatively low (see Appendix A.3), posing a challenge for NAS algorithms.

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
