# OpenReview forum: "A Neural Architecture Dataset for Adversarial Robustness"
_ICLR.cc/2025/Conference — Submitted to ICLR 2025_

### Official Review · Reviewer_gKQB · 2024-10-26

**Soundness:** 3
**Presentation:** 2
**Contribution:** 2
**Rating:** 5
**Confidence:** 2

**Summary:**

The paper presents NARes, a novel dataset of 15,625 unique neural network architectures based on Wide Residual Networks (WRNs), designed to assess adversarial robustness. The dataset includes detailed adversarial training statistics and evaluations against four adversarial attacks, including AutoAttack. The authors claim that NARes offers new insights into adversarial robustness and identifies contradictions in previous studies.

**Strengths:**

1. The initiative to create a comprehensive dataset for adversarial robustness is laudable.
2. The dataset's focus on a wide range of architectures is a valuable contribution to the field.

**Weaknesses:**

1. The paper lacks clarity in explaining key terms and abbreviations, which is detrimental to the readability and accessibility of the research, such as MACs, PGD^20, and PGD-CW^40.
2. The absence of the code and dataset at the time of submission limits the ability of the community to engage with and validate the work.
3. There is a lack of detailed experimental procedures, which is essential for the reproducibility of the findings.
4. The paper would benefit from a clearer structure to better guide the reader through the research and its implications.
5. The establishment of a leaderboard for benchmarking could significantly enhance the utility of the dataset, which is currently missing.

**Questions:**

See weakness.

**Details Of Ethics Concerns:**

No ethical concerns identified upon review. The study appears compliant with standard ethical guidelines.

---

> ### Author Response · Authors · 2024-11-23
> **Official Response to gKQB**
>
> Q1: The paper lacks clarity in explaining key terms and abbreviations, which is detrimental to the readability and accessibility of the research, such as MACs, PGD^20, and PGD-CW^40.
>
> R1: We appreciate you pointing out these writing issues. In the revised version, we explain these terms when they first appeared.
>
> Q2: The absence of the code and dataset at the time of submission limits the ability of the community to engage with and validate the work.
>
> R2: We have uploaded the new supplementary materials with our codebases, including the codes for querying the NARes dataset and estimating NAS algorithms in the paper, as well as the training and evaluation codes for reproduction.
>
> Q3: There is a lack of detailed experimental procedures, which is essential for the reproducibility of the findings.
>
> R3: We have already explained the details in Sec. 3 and Appendix B&C. We also provide the codes for reproducing the generation of NARes in the revision. We are not sure what details you want to know. If you have any questions, please let us know.
>
> Q4: The paper would benefit from a clearer structure to better guide the reader through the research and its implications.
>
> R4: According to the responses of other reviewers, they have a positive attitude towards the structure of the paper. We are not sure which part you are referring to. Nevertheless, in the new revision, we have tried our best to improve the overall quality.
>
> Q5: The establishment of a leaderboard for benchmarking could significantly enhance the utility of the dataset, which is currently missing.
>
> Q5: We highlight that, technically, compared to other similar works, NARes is not a strict benchmark that aims to find the best architectures in a search space for some direct design principles. It is more like a dataset, where the primary objective is to enhance the theories about the relationship between architectures and adversarial robustness rather than directly identifying the best-performing architectures. Besides, as outlined in the limitations section, we do not recommend focusing on identifying the single best architecture individually due to the inherent noise in training and evaluation. Instead, we advocate studying architecture from a distributional perspective. Consequently, a leaderboard for benchmarking, while suitable for other works, is not rational in our context. Lastly, our codebase allows you to query the adversarial robustness of different architectures easily. So, if you are interested in the top architectures, the leaderboard can be easily accessed.

---

> > ### Comment · Reviewer_gKQB · 2024-11-24
> > **Raising Score Following Rebuttal**
> >
> > I appreciate the authors' detailed clarifications. They have thoroughly addressed most of my concerns, and based on their additional explanations and efforts during the rebuttal, I have decided to raise my score from 3 to 5.

---

> > > ### Author Response · Authors · 2024-12-03
> > > **Response to gKQB**
> > >
> > > Dear Reviewer,
> > >
> > > Thank you for your thoughtful and prompt feedback. We truly appreciate the time and effort you have put into reviewing our submission. If there are any additional concerns or questions, we will do our best to provide further clarification.

---

### Official Review · Reviewer_Vzb3 · 2024-10-29

**Soundness:** 3
**Presentation:** 4
**Contribution:** 3
**Rating:** 6
**Confidence:** 4

**Summary:**

The authors propose a new benchmark for architecture search of adversarially robust networks. They evaluate various depth and width settings in WideResNet. They also benchmark various architecture search algorithms on their benchmark.

**Strengths:**

-- This is a well-motivated paper as far as I can tell. Robustness to adversarial attacks is clearly an important problem, to which architecture search is an important component.

-- I find the paper to be sound in terms of discussions of related work. Throughout the paper there is clear references both to the adversarial defense literature and the NAS literature. They also provide clear arguments for the distinctiveness of their method compared to prior NAS benchmarks.

-- The experimental setup appears to be sound and the benchmarking extensive. I find the experiments on the empirical Lipschitz constant to be interesting and against the convention that increasing depth would increase the Lipschitz constant. I believe the authors should consider discussing this connection more, especially in relation to the learning theory literature (for example, the Lipschitz constant can be related to the Frobenius norm of the weight matrices, and in turn related to generalization).

-- The connections between adversarial robustness and the Lipschitz constant are not surprising, but is an important empirical finding especially since the paper studies it systematically for the given architecture family in terms of depth, width. Similarly their findings on depth-width ratio are informative and will be of interest to the NAS community.

**Weaknesses:**

-- The focus on WRN-style architectures, while it makes sense practically, may limit the generalizability of findings to other architectures. For example, aspects like width and depth and their relationship to adversarial robustness may differ across architectures. Vision transformers are a very popular architecture, for example, and one could imagine that they may behave differently.

-- Similarly, the empirical evaluation is only on CIFAR-10. While I understand ImageNet can be expensive (especially for adversarial training), there are several other datasets that are of similar training time e.g. CIFAR-100, TinyImageNet. For this reason and the one above I'm not sure if the benchmark would "catch on".

**Questions:**

-- How do you plan to keep the benchmark relevant as new architectures emerge?

-- What is the roadmap towards new architectures and datasets? Will it be open to the community or do you expect to run them yourselves? Why would someone from the community contribute to it? For example, in typical benchmarks you expect researchers with competing methods to be incentivized to contribute. Who would be incentivized to contribute these kind of extensive NAS comparisons?

Basically I want to know why you expect this benchmark will catch on.

---

> ### Author Response · Authors · 2024-11-23
> **Official Response to Vzb3**
>
> Q1: The focus on WRN-style architectures, while it makes sense practically, may limit the generalizability of findings to other architectures like ViT. Similarly, the empirical evaluation is only on CIFAR-10.
>
> R1: We believe the solution is dependent on how we use NARes. As mentioned in the limitation section, we recommend that the researchers use NARes as the first step for new insights through these off-the-shelf data; then, the generalization or other verifications can be further verified in different settings. Therefore, we think it is fine even if some researchers find some new insights that are not generalized well to other scenarios. Additionally, we added a new experiment in Appendix A.7 that trains a small subset of architectures on Tiny-ImageNet, and the results suggest the generalization of NARes.
>
> Q2: How do you plan to keep the benchmark relevant as new architectures emerge?
>
> R2: In contrast to other similar works, NARes is more like a dataset than serving as a model benchmark. One primary goal of NARes is to provide a high-resolution architecture landscape as an empirical dataset for the theoretical study of architecture and adversarial robustness (AR). We expect it will bridge the gap between theory and practice and refine the fundamental understanding of the relationship between architecture and AR. These fundamental theories will then benefit the improvement of new architectures.
>
> Q3: What is the roadmap towards new architectures and datasets? Will it be open to the community, or do you expect to run them yourselves? Why would someone from the community contribute to it? Who would be incentivized to contribute to this kind of extensive NAS comparisons?
>
> R3: As discussed in R1, NARes is the first step for new research. Considering the training budget, an efficient way is to test a small set of architectures on a new dataset to verify the new ideas. Nevertheless, we will open-source all codes for the process of generating NARes. So researchers with massive resources can continue this work, extending to new architectures and datasets.
>
> For the last two questions, NARes provide another quick evaluation metric for NAS algorithms under the macro search space, which has not been carefully investigated in NAS for AR. On the one side, research of NAS for AR currently focuses more on the topology of the cell-based (micro) search space; on the other hand, many empirical principles and theoretical analyses are conducted on the macro search space, especially on the WRN-style architecture. Therefore, we hope NARes will help bridge the gap between NAS and other AR areas. We have discussed it in Sec. 1. Besides researchers in the NAS community, since we also provide the checkpoints for all architectures during the training, we expect that researchers can utilize these checkpoints to conduct their interested experiments, reducing the training times. For example, researching the robust overfitting issue with respect to the architecture and improving the adversarial training method.

---

> > ### Comment · Reviewer_Vzb3 · 2024-12-02
> >
> > Dear Authors,
> >
> > Thank you for responding. I will retain my score.
> >
> > Best regards,
> >
> > Reviewer Vzb3

---

### Official Review · Reviewer_JLna · 2024-10-31

**Soundness:** 2
**Presentation:** 3
**Contribution:** 2
**Rating:** 5
**Confidence:** 4

**Summary:**

This paper focuses on a valuable topic, namely, the robustness of the models. The main contribution of this paper lies on the proposed NARes dataset. This dataset is constructed for investigating the relationship between the model robustness and model architectures. NARes comprises 15,625 WRN-style architectures adversarially trained and evaluated against various adversarial attacks, such as AutoAttack. The authors provide extensive experimental results and meaningful observations.

**Strengths:**

1. The focused topic is valuable. Model robustness is of great significance in practical deep learning.
2. The scale of the proposed method is quite satisfactory. This paper constructs a large scale dataset that consists of 15,625 different model architectures and 62500 checkpoints in total.
3. The statement is clear and accurate. The organization and the writing of this paper is satisfactory that make the reader easily understand the core idea and meaningful conclusions.

**Weaknesses:**

My biggest concern to this work is its novelty. As far as I know, the RobustArt has not only investigated the relationship between the model architecture and robustness, but also investigated the features among model architectures, the training techniques, the adversarial noise robustness, the natural noise robustness, and the system noise robustness, which in fact takes the first step in this area. Compared with the RobustArt, the strengths of this study could only appear in the dataset scale and fine-grained factors, e.g., the width, depth. #MACs, and LIP of WRNs. I believe this study is meaningful and valuable, but it seems this study is more likely to be a star project in the open-source area. By the way, the statement about the RobustArt in section 2.3 appears to be misleading.

**Questions:**

(1) Plz carefully clarify the differences between this study and the RobustArt in the perspective of novelty.
(2) It could be more acceptable to provide some overall information, such like a leaderboard.
(3) This is quite a good work, but the open-source link is not provided.

---

> ### Author Response · Authors · 2024-11-23
> **Official Response to JLna**
>
> Q1: The paper has a novelty issue compared to RobustArt. Please carefully clarify the differences between this study and the RobustArt from the perspective of novelty.
>
> R1: There are crucial differences between the two works. Different focused perspectives make different experiment strategies. RobustArt serves as a benchmark, providing empirical architecture design principals, while NARes serves as a dataset, providing a high-resolution architecture landscape on adversarial robustness (AR) for theoretical analyses. Our work is closely correlated to the theoretical research about architecture (depth and width) and AR, such as [1-3], which leads to two significant differences:
>
> 1. Like the motivation of NAS-Bench-101 and later series, enumerating the entire search space is a crucial prerequisite to thoroughly analyzing the relationship between architecture and AR. As indicated in Sec.4.3, insufficient data sampling could lead to incorrect or at least inaccurate conclusions. Our exhaustively evaluated architecture dataset with rich diagnostic information provides a reliable data source to verify previous statements and develop new theories.
> 2. RobustArt’s primary source of architectures and corresponding weights are from the BigNAS supernets. However, whether this mechanism could interfere with subnets’ decision boundaries for AR is ambiguous. We have discussed our decision about why we didn’t use this strategy in Appendix B.3. We believe that training architectures independently from scratch, albeit time-consuming, is a practical way to eliminate irrelevant interference and provide reliable data. In addition, we ensure all architectures in our search space have sufficient model capacity for AR demand, resulting in 53.09%~57.39% PGD accuracy.
>
> In summary, NARes is an essential complementarity to previous works, including RobustArt. Besides, when considering the above differences, the most relevant work is [4], and we have discussed its limitations in the paper.
>
> Q2: The statement about the RobustArt in section 2.3 appears to be misleading.
>
> R2: Thank you for pointing out the potential issue with the statement about RobustArt in Section 2.3. We have revised our words in the new manuscript.
>
> Q3: It could be more acceptable to provide some overall information, such as a leaderboard. And the open-source link is not provided.
>
> R3: We upload the codes in the revised supplementary, including the codes for accessing the NARes dataset and the training and evaluation codes for reproduction. These codes will be open-sourced to public once it is accepted. We provide an easy-to-use API to access all the data mentioned in the paper.
>
> Besides, as explained in the limitation section, it is not recommended to study the best architecture individually due to the training and evaluation noise. We recommend studying architectures from a distribution perspective. Moreover, one of our primary goals is to improve our understanding of architecture and AR instead of directly studying the best architecture. Therefore, the leaderboard is unsuitable for our scenario compared to other works. If you are still interested in the best architecture in our search space under some attack metrics, you can easily query it using our codes.
>
> [1] Exploring Architectural Ingredients of Adversarially Robust Deep Neural Networks
>
> [2] A Universal Law of Robustness via Isoperimetry
>
> [3] Robustness in deep learning: The good (width), the bad (depth), and the ugly (initialization)
>
> [4] Robust NAS under adversarial training: benchmark, theory, and beyond

---

> ### Comment · Reviewer_JLna · 2024-11-26
> **Response**
>
> Thanks for your efforts. Some of my concerns are addressed, I will update my rating to 5.

---

> > ### Author Response · Authors · 2024-12-03
> > **Response to JLna**
> >
> > Dear Reviewer,
> >
> > Thank you for your thoughtful and prompt feedback. We truly appreciate the time and effort you have put into reviewing our submission. If there are any additional concerns or questions, we will do our best to provide further clarification.

---

### Official Review · Reviewer_hpsi · 2024-11-09

**Soundness:** 3
**Presentation:** 3
**Contribution:** 2
**Rating:** 6
**Confidence:** 4

**Summary:**

This paper spend tons of GPU resources to construct a huge benchmark on the relation between WRN architecture design and adversarial robustness. Specifically, the authors varied the depths and widths of WRN and used PGD-based adversarial training on CIFAR-10 dataset to get the adversarial robustness of the model architecture.

Several interesting observations are found based on the benchmark results, which may benefit further research. I have concerns on whether those findings generalize to other model architectures beyond WRNs and CIFAR-10.

The benchmark should be very welcomed by the NAS community, as a new benchmark for searching robust model architectures. I'm not sure how the benchmark could help wider communities, since the WRN + CIAFR-10 image classification + PGD adversarial training is only a very narrow scope in terms of robust machine learning. More advanced model architectures, larger-scale datasets, and tasks with better practical applications are missing in the current benchmark design.

**Strengths:**

1. Tons of GPU resources are generously spent to build the new benchmark.

2. The observations from the benchmark results are all very interesting and may inspire future works.

3. The benchmark should be very beneficial to the NAS community to design better search algorithms for robust model architectures.

**Weaknesses:**

My main concern is that the benchmark only covers a tiny scope in the field of robust machine learning.
Specifically, the authors adopted the setting of WRN + CIAFR-10 image classification + PGD adversarial training, without very convincing justification.
I'm aware that many previous adversarial training papers used similar settings. However, as a benchmark paper, it requires more attention, much more than papers proposing new algorithms, when it comes to designing the experiment settings, on which tons of resources will be spent.
In my point of view, the WRN is not the best choice for image classification tasks, or as the backbone of detection and segmentation tasks, in year 2024. For applications running on cloud servers or where computation resources are not a constraint, large transformer-based models are dominant in terms of both clean accuracy and robustness under distributional shifts. For edge-device or other resource constrained devices, efficient vision transformers are dominant (e.g., [1]). Although these models are actually pure CNN models replacing the expensive multi-head self-attention with cheap token mixers like depth-wise convolution, their design of global architectures are very different with WRNs. They use the alternate token-mixer and channel-mixer design inspired by transformers. It is concerning if the conclusions from the benchmark do not generalize to these more advanced architectures.
These transformer and efficient transformer models have also dominated applications such as object detection and segmentation, which have better practical applications than image classification, which leads to my second concern on the choice of the task in the benchmark. I'm fully aware that adversarial training papers typically adopt image classification as the primary benchmark task. But when it comes to real-world applications that requires high-level security, such as autonomous driving and security surveillance, detection and segmentation are more heavily used. As a benchmark paper, it would be good to focus not only on one single task, but on more tasks, especially those with more practical applications. My third concern is that the benchmark only adopts PGD-adversarial training (PGD-AT) as the defense method. It is not clear if the conclusions from PGD-AT generalizes to other defense methods (e.g. TRADE).

[1] FastViT: A Fast Hybrid Vision Transformer using Structural Reparameterization.

**Questions:**

Please see above.

---

> ### Author Response · Authors · 2024-11-23
> **Official Response to hpsi**
>
> Q: The benchmark only includes a narrow scope in robust machine learning from the model architecture (WRN-style), machine learning task (Image Classification), and the adversarial training method (PGD-AT) perspective.
>
> R: We will collectively respond to the weaknesses that the reviewer is concerned about. Besides exploring new architectures, new real-world applications, and new AT methods, another way to contribute to the adversarial robustness (AR) domain is to build its fundamental basis by developing theories and understanding the mechanisms behind it. Both sides are equally crucial.
>
> Technically, compared to other similar works, NARes is not a strict benchmark for models that aims to find the best architectures in a search space. We emphasize that one primary goal of NARes is to provide a high-resolution architecture landscape as an empirical **dataset** for research of the latter mentioned above. Therefore, we also provide rich diagnostic information for every architecture, such as training curves and LIP, which is missing in previous architecture datasets. Based on our observations, we have found some violations against previous theories. Our dataset will help refine the theories, contributing to a more robust understanding of the connection between architecture design and AR. Then, these new theories will help build new algorithms for new architectures, such as ViT, and adapt to more complex tasks.
>
> As a result, our dataset design choice is intended to serve this purpose. The WRN-style model architectures, the image classification task, and the PGD-AT training strategy are applied since they are simple yet effective settings that are widely studied, avoiding introducing unnecessary interference to the relation between architectures and AR, which is important in generating such datasets. Currently, we choose CIFAR-10 as the trade-off between the training budget and the search space size. Although, to our knowledge, NARes is still the most expensive dataset to build in the field. When considering the finite training budget, it is prohibitive to handle all the factors perfectly. Moreover, in the new supplementary, we additionally conduct a new experiment that trains a small subset of architectures on Tiny-ImageNet, and the results in Appendix A.7 suggest the generalization of our observations. Lastly, as mentioned in the limitation section, we recommend that the researchers use NARes as the first step for new insights, benefiting from the off-the-shelf data; then, the generalization or other verifications can be further tested in different settings. Therefore, it is okay even if some new findings are not generalized to other scenarios.
>
> In summary, although NARes is training on a narrow scope, our work provides an essential complementarity of previous architecture datasets and benchmarks and has a unique value in this domain.

---

> > ### Comment · Reviewer_hpsi · 2024-11-30
> >
> > Thank you for your detailed response. The merits you highlighted in your reply were already noted during my initial review and factored into my evaluation score. I agree that the paper, along with the considerable resources dedicated to it, has the potential to benefit the research community. However, my concerns about the conclusions' generalizability, given the dataset's current narrow scope, remain unresolved.
> >
> > That said, I consider this a borderline paper, and opinions on its acceptance or rejection are likely to vary. Taking its merits into account, I have decided to adjust my score from 5 (marginally reject) to 6 (marginally accept). I believe accepting this paper could provide it with more visibility within the community, allowing future researchers to decide whether this setting is worth pursuing further.

---

### Meta-Review · Area_Chair_5gGa · 2024-12-18

**Metareview:**

The paper presents NARes, a large-scale dataset comprising 15,625 WideResNet (WRN)-style architectures designed to explore the relationship between neural architecture and adversarial robustness. While the work addresses an important topic and showcases significant computational effort, it ultimately falls short of acceptance due to limitations in scope, novelty, and practical impact.

The primary contribution of this work lies in the creation of NARes, which includes adversarially trained architectures evaluated against various attacks, such as AutoAttack, and provides detailed statistics like Lipschitz constants and stable accuracy metrics. The dataset aims to facilitate research on adversarial robustness by offering a comprehensive and fine-grained resource. The authors claim that NARes can help researchers derive new insights and verify existing theories in this domain.

The strengths of the paper include its focus on an important topic, adversarial robustness, which is critical for real-world machine learning applications. The extensive computational effort behind training and evaluating a large number of architectures is commendable and demonstrates a commitment to thorough empirical evaluation. Furthermore, the fine-grained dataset and the rich diagnostic information provided for each architecture could inspire further studies and theoretical advancements in the field. The paper is also well-structured, making it easy for readers to follow its main contributions and conclusions.

However, the paper has notable weaknesses that undermine its overall impact. First, the scope of NARes is narrow, as it focuses exclusively on WRN architectures trained on CIFAR-10 using PGD-based adversarial training. This limited focus raises questions about the generalizability of its findings to other architectures, datasets, and tasks. For example, modern architectures like vision transformers (ViTs) dominate many applications and may exhibit significantly different robustness characteristics. Additionally, CIFAR-10, while computationally convenient, is insufficient for assessing real-world applicability, and alternative datasets such as CIFAR-100 or TinyImageNet could have provided more diversity. Second, the paper’s novelty is limited compared to existing works like RobustArt, which explores similar topics but with broader architectural and methodological scope. While NARes provides finer-grained data and a larger scale, it does not introduce fundamentally new ideas or methodologies that significantly advance the field.

The practical utility of the dataset also remains uncertain. The absence of a leaderboard or clear benchmarks reduces its immediate appeal to the community, and the dataset’s availability only after acceptance limits engagement and validation opportunities. Furthermore, the authors do not convincingly justify why WRN-style architectures and the specific adversarial training method used were chosen, other than their simplicity. This choice could limit the dataset’s applicability to a broader range of research contexts, such as real-world tasks like object detection or segmentation, where robustness is crucial. The authors' response that NARes serves as a foundational dataset rather than a direct benchmark does not fully address these concerns, as it still restricts its relevance and adoption in practical scenarios.

In summary, while the paper demonstrates commendable effort in dataset construction and provides a potentially useful resource for certain niche applications, it fails to meet the standards of generality, novelty, and utility required for acceptance at a top-tier venue. The dataset’s narrow focus, limited generalizability, and incremental contributions relative to existing works do not justify its inclusion. To address these issues, the authors would need to extend the scope of their analysis to include more diverse architectures, datasets, and tasks, as well as provide more compelling evidence of NARes’s broader impact. Thus, the recommendation for this submission is rejection.

**Additional Comments On Reviewer Discussion:**

During the rebuttal period, the reviewers raised several critical points regarding the paper's scope, novelty, and practical impact, alongside questions about its generalizability and reproducibility. These concerns shaped the subsequent clarifications and changes offered by the authors, which were weighed carefully in the final decision to reject the paper.

One major point raised was the narrow focus of the dataset, which exclusively featured WRN-style architectures trained on CIFAR-10 with PGD-based adversarial training. Multiple reviewers expressed skepticism about the generalizability of the findings to other architectures, such as vision transformers (ViTs), or to other datasets and tasks, including detection or segmentation. While the authors argued that their work aimed to provide a foundational resource for understanding architecture-level adversarial robustness, they failed to sufficiently justify why WRN and CIFAR-10 were chosen beyond computational convenience. They introduced a small experiment on TinyImageNet in the supplementary materials to demonstrate the potential for generalization but admitted the dataset’s limitations in the manuscript. While this additional data helped to a limited extent, it did not adequately address concerns about the scope of NARes, especially for a benchmark paper expected to have broader utility.

The novelty of the work compared to existing resources, such as RobustArt, was another central issue. Reviewers noted that the submission primarily extended the scale and granularity of previous studies but did not provide fundamentally new insights or methods. The authors responded by emphasizing differences in focus: RobustArt acts as a benchmark for empirical architecture design principles, while NARes provides a high-resolution dataset for theoretical exploration. However, reviewers found this distinction insufficient to address concerns, as both works share significant conceptual overlap, and NARes does not introduce clear innovations that extend the field meaningfully. Additionally, claims of new insights contradicting prior works were not substantiated with enough evidence to sway the reviewers.

Reproducibility and practical utility were also flagged. Reviewers criticized the lack of accessible open-source code and dataset at the time of submission, as this significantly limited the community's ability to validate or build upon the work. In their rebuttal, the authors uploaded supplementary materials, including the dataset querying code and experimental setups, while committing to public release upon acceptance. They also clarified their decision not to include a leaderboard, arguing that NARes was intended as a dataset for theoretical study rather than a direct performance benchmark. However, the reviewers remained unconvinced that this design choice was appropriate for a community resource, particularly given the dataset’s potential as a benchmarking tool for adversarial robustness.

Several reviewers raised concerns about the clarity of the paper and its technical presentation. Issues such as unexplained abbreviations (e.g., MACs, PGD variants) and unclear experimental procedures were highlighted as detrimental to readability and reproducibility. The authors revised their manuscript to address these points, adding clarifications and improving explanations for key terms. While this was a positive step, it was considered incremental and did not significantly alter the reviewers’ overall assessment of the submission.

In weighing the rebuttal and the authors’ responses, the decision ultimately rested on the fundamental limitations of the work. While the dataset was recognized as a potentially valuable resource for a niche community, its narrow focus, limited novelty, and unresolved generalizability issues detracted from its broader impact. The additional experiments and clarifications during the rebuttal addressed some specific concerns but failed to sufficiently mitigate the key issues raised. Specifically, the lack of diversity in architectures and datasets, along with the absence of compelling evidence for the dataset’s general applicability, meant that the paper did not meet the standards expected for a major conference. Furthermore, the reliance on computational resources alone, without strong methodological or conceptual contributions, reinforced the perception that the work was incremental.

In conclusion, while the rebuttal period demonstrated the authors’ willingness to engage with feedback and make improvements, the responses did not sufficiently address the core weaknesses identified by the reviewers. The paper’s contributions remain limited to a specific setting, and its broader relevance to the adversarial robustness community remains uncertain. As such, the final decision was to reject the submission.

---

### Decision · Program_Chairs · 2025-01-22

Reject